# Evidences from Rewarding System, FRN and P300 Effect in Internet-Addiction in Young People SHORT TITLE: Rewarding System and EEG in Internet-Addiction

**DOI:** 10.3390/brainsci7070081

**Published:** 2017-07-12

**Authors:** Michela Balconi, Irene Venturella, Roberta Finocchiaro

**Affiliations:** 1Research Unit in Affective and Social Neuroscience, Department of Psychology, Catholic University of the Sacred Heart, 20123 Milan, Italy; irene.venturella@unicatt.it (I.V.); roberta.finocchiaro@unicatt.it (R.F.); 2Department of Psychology, Catholic University of the Sacred Heart, Milan Largo Gemelli, 1, 20123 Milan, Italy

**Keywords:** Internet addiction, IAT, FRN, P300, BAS, reward bias, attention

## Abstract

The present research explored rewarding bias and attentional deficits in Internet addiction (IA) based on the IAT (Internet Addiction Test) construct, during an attentional inhibitory task (Go/NoGo task). Event-related Potentials (ERPs) effects (Feedback Related Negativity (FRN) and P300) were monitored in concomitance with Behavioral Activation System (BAS) modulation. High-IAT young participants showed specific responses to IA-related cues (videos representing online gambling and videogames) in terms of cognitive performance (decreased Response Times, RTs; and Error Rates, ERs) and ERPs modulation (decreased FRN and increased P300). Consistent reward and attentional biases was adduced to explain the cognitive “gain” effect and the anomalous response in terms of both feedback behavior (FRN) and attentional (P300) mechanisms in high-IAT. In addition, BAS and BAS-Reward subscales measures were correlated with both IAT and ERPs variations. Therefore, high sensitivity to IAT may be considered as a marker of dysfunctional reward processing (reduction of monitoring) and cognitive control (higher attentional values) for specific IA-related cues. More generally, a direct relationship among reward-related behavior, Internet addiction and BAS attitude was suggested.

## 1. Introduction

Internet addiction (IA) was classified as one category of behavioral addiction, representing a specific impairment that involves online and/or offline web misuse, and it is mainly relevant for young generations [1,2,3]. It was considered as an impulse control disorder [4,5,6] with significant impairment of relevant executive functions [7]. A second main aspect implicated in IA is a deficit in the rewarding mechanism, since it was shown to induce a “reward bias” for potential rewarding cues, such as videogames or web gambling stimuli [8,9,10]. In addition, the main components of reward sensitivity, executive deficits and impulsiveness are supposed to have an important role in explaining the IA [11].

In previous research the existence of a strong relationship between limited impulse control and addictive behaviors was first observed, such as pathological gambling, substance and alcohol abuse [12,13,14,15,16]. It was also reported that subjects with IA were more impulsive than were controls as measured by the Barratt Impulsiveness Scale-11 (BIS-11, [17]), the Go-Stop impulsivity paradigm and response inhibition paradigm [18]. Secondly, it was shown that impaired working memory can lead to limited decision-making capacity, which induces inability to plan an adequate best long-term strategy and to inhibit immediate reward-seeking [19,20,21].The response inhibition, as assessed through Go/NoGo tasks, can be defined as the act of withholding or terminating a behavioral response and is considered to be governed by a cognitive inhibitory process [22].

Thirdly, it was revealed that reward motivation significantly correlates with drug addiction, specifically for young people [23,24].The reward deficit syndrome was proposed as a possible contributing factor to the development of substance abuse disorders [18], since addiction may be related to greater receptiveness to the reinforcing effect of drugs and other similar rewarding stimuli [16,22]. Also for IA this dysfunction was observed in recent research (see [6,7] for a complete review).

Regarding the cortical correlates of addictive behavior, recent studies found the involvement of the prefrontal cortex (PFC) through its regulation of the limbic reward regions as well as its involvement in higher-order executive functions [14,23,25,26,27,28]. Specifically three main effects were found. Firstly, it was observed hyperactivity in the emotional system, mediated by frontal and medial structures, such as the orbitofrontal cortex (OFC), but also anterior cingulate cortex (ACC) and amygdala, which exaggerate the rewarding impact of reinforcing cues. Secondly, it was also found anomalous cortical responsiveness in dorsolateral prefrontal cortex (DLPFC), which was found to predict the long-term consequences of a given action [19,25,29]. Thirdly it was underlined a dysfunction in the dopaminergic mesolimbic reward system which is suggested to support attention allocation for dependence-associated cues. Indeed these cues are made exaggeratedly salient, as reported in substance abusers and impulsive individuals [24,30,31,32]. More specifically PFC was implicated in rewarding bias, and, whereas the left PFC was shown to be more implicated in approach-related and rewarding conditions, the right PFC was found to be more involved in withdrawal-related motivations and inhibitory mechanisms [33,34,35]. Both approach and withdrawal motivations are paralleled by the reward and punishment contingencies, as shown in recent study on EEG (electroencephalographic) measures [36,37,38]. More specifically, BIS (Behavioral Inhibition System) and BAS (Behavioral Activation System) measure represents an usable tool to test this reward-sensitivity and some disfunctional aspects [10,36,37,39,40,41,42,43,44]. Both approach- and withdrawal-motivations are paralleled by the reward and punishment effects. High insensitivity for punishment together with a strong reward dependence results in a disadvantageous pattern of decision making. Indeed it was deminstrated that sensitivity to punishment moderated the effect of sensitivity to reward to predict the likelihood of having any gambling problems [45]. Overall, previous results suggest that individual differences in sensitivity to punishment and sensitivity to reward are functionally associated with gambling problems.

Therefore, the role of the reward system, from, one hand, and of the attentional bias and impulse control, from the other, were supposed to explain and elucidate these anomalous mechanisms in the case of IA. Specifically, it was demonstrated that ventromedial prefrontal cortex (VMPFC) is a key structure in decisional processes, depending on the integrity of two sets of neural systems: the first one is critical for the working memory and the related executive functions (such as inhibition, planning, and cognitive flexibility), which includes DLPFC; the second one is critical for processing emotional and motivational information related to reward, in which more medial structures (such as insular cortex and cingulate cortex) are relevant [19].

However, little is known about individual differences in reward mechanisms and executive functions, mediated by frontal system, in the case of internet-addiction and decisional processes related to IA. More generally, previous studies using EEG, and specifically time-frequency analysis related to the event-related potentials in response to target detection, have found significant differences between different types of addictions and control subjects in decisional processes. More generally brain oscillations were used to explore brain correlates of different types of addiction, although only in a limited number of cases focusing on an ample range of brain oscillations [24,25,46] or they were not specifically related to internet-addiction [47].

Recently some research focused on a specific event-related potentials (ERP) effect, the feedback negativity (FRN) to uncover the neurocognitive correlates of decisional behavior in case of dysfunctional conditions, as addiction. This ERP effect is a typical mediofrontal negativity, peaking at about 200–350 ms after the onset of the feedback stimulus, that signals an unfavorable compared to favorable outcome [34,48,49,50]. Morevoer, it is involved in performance minotoring and it was observed that it is probably cortically generated near the MFC, mainly the anterior cingulate cortex (ACC) [51]. In addition, it was supposed that the processing underlying the FRN are triggered by phasic dopaminergic signals, coding reward prediction error. These prediction error signals is successvely conveyed to the ACC where they lead to adjustements in subsequent action selection [52]. Althought different results were found about the significance of FRN [53], this ERP effect is particularly adapt to analyze the outcome expectancies and the eventual deficit in feedback control mechanisms, as supposed in anomalous rewarding effect, such as perception of increasing or decreasing of rewarding power of responses.

In fact, the absence or anomalous functioning of the reward prediction error mechanism (for example induced by an anomalous increasing of sensitivity to rewarding cues) should induce a significant and systematic reduction in the FRN amplitude for highly rewarding cues and, in contrast, a systematic increasing for low rewarding cues (such as more neutral stimuli). This insensitivity or an equal reward prediction error should be related to the inability to attribute an adequate relevance to some responses (to the “neutral” stimuli) which are represented as an “erroneous” behavior, from one hand; and with an exaggeration of the rewarding effect for addiction-related cues, with a consequent impact on the general decisional performance (an inadequate attentional distribution to the rewarding/non rewarding stimuli), from the other hand. That is, a sovraestimation of the rewarding effect in case of the appearance of addiction related-cues increases the dysfunctional behavior [54,55]. Similar deficit in error-related effects were found also in other research who considered Error-Related Negativiy (ERN), with a consistent reduction of ERN for higher IAT [7] .

A second relevant ERP deflection is the P300, peaking around 300–600 ms after stimulus onset at posterior recording sites. It was previously used to explore the impariment of the executive functions in decisional processes, that is the difficulty in updating the incoming contextual information. Morevoer, P300 has been shown to be sensitive to the significance and occurrence probability of a stimulus [55,56] as well as task complexity [57]. The increasing amplitude of this positive deflection was observed to represent the necessity to restore adjunctive information to updating the context [1,48,58] or when a more salient and relevant event is observed which is able to produce an automatic attentional response. Thus, it was found that more relevant outcomes automatically ingenerate an increased P300. In recent study, P300 amplitude was specifically observed to be increased in the case of an inhibitory task (Go/NoGo task), as a marker of inhibitory deficit for IA [1].

Therefore, taken together these two ERP measures are used as marker of the increased inability to adopt an adequate cognitive strategy in response to a decisional context and the presence of some rewarding bias, in concomitance to some anomalous automatic attentional responses. In other words, these ERP effects should be explained by a bias in reward sensitivity and a concomitant deficit in attentional behavior.

However, whether and how IA is related to rewarding mechanisms in response to Go/NoGo from one hand; and how attentional functions deficits are related to in IA on the other hand, is actually unexplained . In addition, no previous study direclty considered the significance of Carver and White’s BIS/BAS measures for IA, by comparing the high- vs. low-BAS construct (and specifically BAS-Reward subscale) with specific measures of web addiction and brain activity in the case of rewarding. In addition, no previous research monitored these ERP measures to furnish a complete overview of rewarding deficits.

To test the rewarding bias and attentional deficits based on IA construct, in the present research attentional inhibitory task (Go/NoGo task) was performed. Internet Addiction Inventory (IAT, [59]) was applied to distinguish between high- or low-IAT profiles and to test its effect during the performance in response to specific potentially rewarding cues since internet-related (videos representing online gambling and videogames) or neutral contexts (as normal sport game). Compared with other measures, IAT was applied to an ample sample to select subjects who meet specific criteria. In addition, it was created according to the diagnostic criteria of the DSM-IV for pathological gambling and it was adapted for the diagnosis of Internet Addiction. Finally, IAT’s easiness and self-administration make it a highly usable and feasible tool to measure pathological gambling.

Thus, we expected that more high-IAT (with pathological profiles) should show a reduced FRN and a higher P300 for reward-related cues compared to non-rewarding related cues, due to the exagerated rewarding power and attentional allocation to reward-related conditions compared to no reward-related conditions. These effects should be mainly reported in NoGo task, when subjects have to inhibit their response to the external cues and they have to monitor their attentional behavior.

In addiction, taking into account the BAS contribution, higher-BAS subjects should show the inability to control this anomalous monitoring behavior (with a costant reduced FRN and a systematic P300 increasing) when the rewarding stimuli are presented [10]. This fact should be explained taking into account the significance of BAS for the the rewarding construct, with a potential anomalous rewarding sensitivity in response to specific stimulus category (gamblig stimuli and videogames) for higher-BAS. Finally, a direct association should be observed between IAT and BAS constructs.

## 2. Methods

### 2.1. Participants

Twenty-four young volunteers took part in the study (M (Mean) = 25.09, Sd (Standard deviation) = 1.03; age range = 20–27, 13 women). All subjects were undergraduate students at the Catholic University of Milan and were right-handed, with normal or corrected-to-normal visual acuity. Exclusion criteria were history of psychopathology not related to Internet addiction for the subjects or immediate family members. No specific neurological or psychiatric pathologies were observed by clinical colloquium. Other addictive behaviors were excluded from the sample. A specific questionnaire was submitted to explore the drug and Internet use by the subjects. The selected subjects met with the inclusion criteria (see the following for the IAT parameters). All participants gave informed written consent for participating in the study, and the research was approved by the Ethical Committee (Department of Psychology) of the institution where the work was carried out.

### 2.2. Procedure

The participants sat on a comfortable chair in front of a Pc screen (1280/1024 pixel). The Pc was placed approximately 60 cm from the subject, with a visual horizontal angle of 4° and a vertical angle of 6°. The Go/No-Go task was a modified version of the experimental task used by Petit and colleagues (2012) (see [8] for this version) and it was composed of four blocks of 120 stimuli per each, which were divided in 84 Go trials and 36 No-Go trials for each session. The blocks consisted of randomized presentation of background pictures, appearing at the center of the screen: gambling (G), videogames (VG) and neutral (N) for 500 ms. Successively the letter M or W appeared in the center of this background picture for 200 ms, and then the initial background picture came back for 1300 ms (Figure 1). The letters were presented in a random order to ensure the same amount as a percentage of the trials Go (70%) and No-Go (30%) for each block and category. Participants were required to press a button as fast as possible when they saw the Go stimulus and to withhold the response for the No-Go stimulus. They had a maximum of 1500 ms to press the button before the next letter appeared. Moreover, they were asked to reduce moving and blinking during the task in order to control EEG artifacts during registration. Each participant completed a total of 480 trials. In order to familiarize with the task, the participants completed a short session of 20 trials (70% Go and 30% NoGo) on a black background. After the Go/No-Go task, the participants were submitted to a debriefing phase and to the post-evaluation questionnaires (State Anxiety Inventory (STAI-Y); Beck Depression Inventory (BDI-II); Behavioral Activation System (BAS)). In contrast IAT was administered before the experimental session

### 2.3. Stimuli

In the experimental task the stimuli were two capital white letters (M and W; size of 500 × 400 mm) in Times New Roman font and background pictures (gambling-related, videogames-related and neutral contexts) (Figure 1) displayed on a 15-inch monitor. 20 voluntaries, matched with age and sex with the experimental group, evaluated these pictures for gambling- and videogames-related context, considering four dimensions: relevance, familiarity, valence and arousing power (for this procedure see [8]). 18 pictures were selected and categorized into three types: 6 neutral stimuli; 6 gambling-related stimuli; 6 pictures for video games-related condition.

### 2.4. IAT Scores

Internet Addiction Test (IAT) [59] was applied to an ample sample to select subjects who meet specific criteria. IAT was created according to the diagnostic criteria of the DSM-IV for pathological gambling and it was adapted for the diagnosis of Internet Addiction. The questionnaire consists of 20 items measured with four-points Likert scale (ranging from “never” to “always”). The score was valued according to the cut-off: score between 0 and 30 (none): Internet usage below the average; score between 31 and 49 (mild): an average Internet user, which can sometimes happen to surf the net a bit too long but without losing control of the situation; score between 50 and 79 (moderate): the person already has several problems because of the Internet and it should reflect on the impact these issues have on his life; score between 80 and 100 (severe): the use of the Internet is excessive and is causing considerable problems to the person. Two sub-groups of subjects were created based on this total score: high-IAT with score more than 80 (final N (total number) = 12, M = 82.97; Sd = 5.43); low-IAT with score less than 40 (final N = 12, M = 38.34; Sd = 5.03). Gender was balanced across-group. The Cronbach’s alpha coefficient was from 0.81 to 0.90 (48.11).

### 2.5. BIS/BAS Scores

BIS and BAS scores were calculated for each subject by using the Italian version of Carver and White Questionnaire (1994) [60]. It included 24 items (20 score-items and 4 fillers, each measured on four-point Likert scale), and two total scores for BIS (range = 7–28; items 7) and BAS (range = 13–52; items 13). BAS also includes three subscales (Reward, 5 items, Drive, 4 items, and Fun Seeking, 4 items). The questionnaire was submitted to the subject after completing the experimental phase. Based on these measures, two total scores (BIS and BAS total) and three BAS subscale scores were calculated. The mean values and standard deviations for each scale were respectively for BIS: 18.93(2.77); BAS: 39.11(1.22); Reward: 16.55(1.77); Drive: 14.98(1.99); Fun Seeking: 12.39(2.61). Finally, Cronbach’s alpha was calculated for BIS (0.93) and BAS (0.89) and separately for each BAS subscale (Reward 0.90; Drive 0.89, and Fun Seeking 0.93).

### 2.6. EEG Recordings and Data Reduction

EEG recordings were performed with a 32-channel DC amplifier (SYNAMPS system) and acquisition software (NEUROSCAN 4.2, V-AMP: Brain Products, München, Germany. Truscan: Deymed Diagnostic, Hronov, Czech) during task execution. An ElectroCap with Ag/AgCl electrodes was used to record EEGs from active scalp sites referred to the earlobes (10/20 system of electrode placement [61]. Data were acquired using a sampling rate of 500 Hz, with a frequency band of 0.01 to 50 Hz. An off-line common average reference was successively computed to limit the problems associated with the signal-to-noise ratio [62]. Additionally, two EOG electrodes were sited on the outer canthi to detect eye movements. The impedance of the recording electrodes was monitored for each subject prior to data collection and was always below 5 kΩ. After performing EOG correction and visual inspection, only artifact-free trials were considered (rejected epochs, 2%). The signal was visually scored, and portion of the data that contained artifacts were removed to increase specificity. Blinks were also visually monitored. Ocular artifacts (eye movements and blinks) were corrected using an eye-movement correction algorithm that employs a regression analysis in combination with artifact averaging [63]. This selected procedure used to artifacts rejections was proven to be effective to eliminate the noise from the signal without an excessive “smoothing” of the data and without eliminating too much information. An averaged waveform (off-line) was obtained for each experimental condition. The peak amplitude was quantified relative to the 100 ms pre-stimulus, and the onset was coincident with the appearance of the letter on the monitor [64], taking into account the most negative peak and positive value within the two temporal windows respectively of 150–300 and 300–400 ms post-stimulus.

Distinct peak profiles were calculated respectively for Go/NoGo condition and each stimulus category, and distinct analyses were applied to each the average profiles. Subsequently, localization (four sites: frontal, central, temporo-parietal, and occipital) and lateralization (three sides: left, central, and right) factors were considered in applying statistical analysis. Specifically, we measured left, central and right frontal (F3, Fz, F4), middle-central (Cz, C3, C4), temporo-parietal (P3/T7, Pz, P4/T8; the left and right localizations were obtained as the mean value of parietal and temporal sites) and occipital (Oz, O1, O2) brain activity. The mean latencies of the two deflections were approximately 230 and 330 ms, and they did not vary across the experimental conditions.

## 3. Data Analysis

The statistical analyses were subdivided in two steps. A first set of ANOVAs, applied respectively to the dependent measures of ERs, RTs, FRN and P300, in response to Go/NoGo task and to different stimulus condition (videogames; gambling; neutral). A second set of correlational analyses finalized to explore the relationship between BAS (and BAS-subscales) and IAT measure; BAS and ERPs measures.

### 3.1. ERs

The behavioral measures of ERs (number of errors out of the total of trials) were subjected to a three-way repeated measures ANOVA, with between-subject IAT (2), and the within-subjects factors Condition (2, Go-NoGo) and Stimuli (3). Errors associated with inhomogeneity of variance were controlled by decreasing the degrees of freedom using the Greenhouse-Geiser epsilon. Post-hoc analysis (contrast analysis for ANOVA, with Bonferroni corrections for multiple comparisons) was applied in case of significant effects. Significant effects were found for IAT × Condition × Stimuli (*F*(2,46) = 8.34, *P* = 0.001, η^2^ = 0.36) (Figure 2a). As revealed by simple effects ERs decreased for videogames (*F*(1,23) = 8.87, *P* = 0.001, η^2^ = 0.37) and gambling cues (*F*(1,23) = 8.16, *P* = 0.001, η^2^ = 0.36) in Go for high-IAT more than low-IAT. Similarly ERs decreased for videogames (*F*(1,23) = 8.50, *P* = 0.001, η^2^ = 0.36) and gambling stimuli (*F*(1,23) = 9.06, *P* = 0.001, η^2^ = 0.39) in NoGo condition for high-IAT more than low-IAT.

### 3.2. RTs

RTs were subjected to a three-way repeated measures ANOVA (IAT × Condition × Stimuli). Significant effects were found for IAT × Condition × Stimuli (*F*(2,46) = 10.07, *P* = 0.001, η^2^ = 0.40) (Figure 2b). Simple effects revealed lower RTs for videogames (*F*(1,24) = 9.51, *P* = 0.001, η^2^ = 0.36) and gambling stimuli (*F*(1,24) = 7.56, *P* = 0.001, η^2^ = 0.30) in NoGo for high-IAT more than low-IAT.

### 3.3. ERP Data

Morphological analysis of ERPs showed two significant negative deflections within the 150–300 and 300–400 ms temporal window. The ERP data were subjected to a four-way mixed-design ANOVA, in which the between-subjects group IAT (2) and within-subjects Condition (2), Stimuli (3), Lateralization (3), Localization (4) factors were applied to the peak amplitude variable. Localization (four sites: frontal, central, temporo-parietal, and occipital) and Lateralization (three: left, central, and right) were calculated. Specifically, we measured left, central and right frontal (F3, Fz, F4), middle-central (Cz, C3, C4), temporo-parietal (P3/T7, Pz, P4/T8) and occipital (Oz, O1, O2) brain activity.

### 3.4. FRN

Significant main effects were found for Localization (*F*(3,23) = 7.99, *P* = 0.001, η^2^ = 0.35), and Condition × IAT (*F*(2,23) = 7.37, *P* = 0.001, η^2^ = 0.32). The other main or interaction effects were not statistically significant. The FRN effect was increased in temporo-parietal and occipital areas than the other areas (for all comparisons *P* < 0.01). About the interaction effect, decreased peak amplitude was found for high-IAT than low-IAT in response to gambling and videos for both Go (respectively (*F*(1,23) = 9.11, *P* = 0.001, η^2^ = 0.39); (*F*(1,23) = 11.10, *P* = 0.10, η^2^ = 0.43) and NoGo condition (*F*(1,23) = 9.60, *P* = 0.001, η^2^ = 0.40) (*F*(1,21) = 7.13, *P* = 0.001, η^2^ = 0.33) (Figure 3). Moreover, in high-IAT a lower amplitude of FRN was found in response to gambling and videos compared to neutral stimuli for both Go (respectively (*F*(1,23) = 9.31, *P* = 0.001, η^2^ = 0.38; (*F*(1,21) = 9.02, *P* = 0.001, η^2^ = 0.39) and NoGo (*F*(1,23) = 8.67, *P* = 0.001, η^2^ = 0.36; (*F*(1,23) = 9.45, *P* = 0.001, η^2^ = 0.38) condition.

### 3.5. P300

Significant main effects were found for Localization (*F*(3,92) = 8.16, *P* = 0.001, η^2^ = 0.36), and Condition × IAT (*F*(1,23) = 8.11, *P* = 0.001, η^2^ = 0.35). On the contrary, the other main or interaction effects were not statistically significant. The P300 effect was mainly increased in temporo-parietal than frontal, central and occipital areas (for all comparisons *P* < 0.01). Moreover, increased peak amplitude was found for high-IAT than low-IAT in response to gambling and videos in NoGo condition (respectively (*F*(1,23) = 7.53, *P* = 0.001, η^2^ = 0.33); (*F*(1,23) = 7.13, *P* = 0.001, η^2^ = 0.32) (Figure 4). Moreover, high-IAT showed a higher P300 amplitude in response to gambling (*F*(1,23) = 9.61, *P* = 0.001, η^2^ = 0.39) and videos (*F*(1,23) = 8.70, *P* = 0.001, η^2^ = 0.37) comparing NoGo to Go condition.

### 3.6. Correlational Analysis

Pearson’s correlation analysis (across-subject correlations) was applied to BAS (and BAS-subscales), IAT and FRN/P300 measures (Figure 4a–f). There was a significant positive correlation between BAS and IAT (*r* = 0.602; *p* < 0.001). In addition, BAS-Reward subscale was correlated with IAT (*r* = 0.596; *p* < 0.001). BAS and BAS-Reward subscale were also significantly negatively correlated to FRN (*r* = −0.596; *p* < 0.001; *r* = −0.511; *p* < 0.001) and positively correlated to P300 (*r* = 0.591; *p* < 0.001; *r* = 0.497; *p* < 0.001) amplitude. No other correlational value was significant.

## 4. Discussion

The present research aimed to explore the role of the rewarding mechanisms and attentional functions deficits in IA in a sample of young people. IAT, ERPs (FRN; P300) and BAS were used as integrated measures to test behavioral response and brain activity toward potential Internet addiction-stimuli, such as gambling and videogames, compared to neutral cues when a Go/NoGo task was submitted. Three main effects were found. Firstly, IAT affected the subjective responses to more rewarding cues, with increased performance (reduced ERs and RTs) for high-IAT. Specifically in both Go and NoGo condition high-IAT subjects revealed decreased ERs and RTs compared with low-IAT when they have to respond to rewarding cues. Secondly, both the FRN and the P300 effects were modulated by IAT. Thus, for high-IAT, FRN amplitude decreased and P300 amplitude increased in response to high-rewarding cues. Finally, BAS measure, and specifically BAS-reward subscale, was related to IAT and to ERPs variations.

### 4.1. Cognitive Effects, Rewarding and Attentional Bias in IA

Firstly a main effect was found in relationship with IAT construct, since subjects rated as higher in IAT adopted a specific behavior in response to Go-NoGo task in relationship with the stimulus category. Indeed they demonstrated to be more significantly responsive to potentially rewarding conditions, i.e., videogames and gambling cues, with a general higher cognitive performance. Specifically, the performance (ERs) was affected by stimulus type: indeed videogames and gambling stimuli registered the lowest ERs values. In this case, a sort of “facilitation effect”, with an increased performance for more salient stimuli, may be suggested. Therefore the subjective performance may present a more “immediate” and “impulsive” response, and, in concomitance, a better outcomes for the most salient category (gambling and videogames). A similar effect was found for RTs, with decreased RTs in response to rewarding-cues but only for NoGo condition. Therefore, the suggested facilitation effect was observable exclusively when the inhibition was more salient (NoGo) than in Go condition. These results are in line with the supposition that addicted individuals commonly exhibit a decreased ability to control the desire to obtain desired things such as drugs (i.e., deficit in inhibitory control), despite knowledge about the aversive consequences following drug intake or the low expectation of actual pleasure expected from the drug (i.e., decision making and reward consequences) [8,65]. These results may also suggest a significant “rewarding effect” which was able to significantly improve the subjective performance and to empower the cognitive outcomes [66].

In fact, taking into account both the ERs and RTs effects, we may also suppose that higher IAT scores are paralleled by an attentional bias, with a direct facilitation to find the rewarding cues and, therefore, a significant reduction of ERs. This effect appears also potentiated for RTs measure, in the case of the NoGo condition, where the subjects have to activate the inhibition skills in relationship with the task (to not respond). Therefore more than a simple “impulsivity behavior” this result could be in line with an effective rewarding bias, which is able to induce a better cognitive performance (i.e., an attentional facilitation) for the specific rewarding cues.

### 4.2. ERPs Effects: FRN and P300

This explanation may also describe the two ERPs modifications we found. Indeed, from one hand, FRN showed a significant distinct profile in response to specific stimulus category for high-IAT: gambling and videogames induced a relevant reduction of FRN amplitude compared to more neutral stimuli. This fact may be in favor of a sort of “attenuation effect” of the feedback control mechanisms for stimuli perceived as immediately rewarding and attended; whereas the converse relevant increased peak amplitude of FRN for more neutral stimuli may suggest a sort of a perceived “error feedback” in their responses related to a less relevant category, more unexpected and less preferred.

The observed FRN profile may be compared with previous research that used brain oscillations (mainly low-frequency bands). Indeed it was found that deficits in control-related processes such as behavioral feedback might be related to rewarding bias. In addition, it was shown that this band modulation depends on activity of motivational systems and participates in salience detection [23]. More generally, althought different results were found about the significance of FRN, this ERP effect is particularly adapt to analyze the eventual deficit in feedback control mechanisms, as supposed in “gambling behavior” or dependence behavior. In fact, the absence, or anomalous functioning, of the reward prediction mechanism should induce a significant reduction in the FRN amplitude when subjects process rewarding stimuli. Moreover, a sovraestimation of the potential gain in case of (apparent) reward condition could evidence the dysfunctional effect of IA [54]. Similar effect, with impaired error-related behavior, were found by previous reserch that monitored ERN [12], with reduced ERN profile for high IAT.

In concomitance, the modulation of the P300 attentional marker may have signaled the imminent relevance of the rewarding category in comparison with the no-rewarding category for high-IAT. Indeed, the observation of slightly larger P300 amplitudes after rewarding-related might index stimulus salience [53,67]. From another perspective, these results could suggest a general modulation of the attentional and executive functions in updating the internal and external representation, not responding in an equivalent manner to each stimulus category. However, it should be noted that this increased P300 amplitude for high-IAT was present mainly in response to NoGo condition, where subjects had to inhibit their response. Therefore, the significant impact of such categories (videogames and gambling) could reveal also the necessity for the subjects to highly control and suppress their behavior in response to specific, more “sensitive” to IA and potentially “rewarding” categories compared with neutral ones. The cognitive costs of this control behavior may be well represented by the selective increasing of P300 only for the high-IAT.

Therefore, the P300 effect could integrate and reinforce what we observed for the FRN effect: the improved “attentive” response toward the rewarding cues and a higher more cognitive cost to activate the suppression of their response in high-IAT, as revealed by higher P300, may have contributed to attenuate the feedback control (lower FRN amplitude), with limited activation of feedback mechanisms in the case of gambling/videogames, in comparison to neutral conditions.

### 4.3. BAS and BAS-Reward Subscale Contribution

An adjunctive effect was observed about the relationship between IA and BAS construct, from one hand; between BAS and FRN/P300, from the other hand. Indeed firstly we observed a strength correlation between higher IAT and higher BAS profile, mainly in response to rewarding-subscale. These results partially confirmed what found in previous research with drug addiction. In fact, a strong relationship was also shown between impulsivity, drug-dependence and BAS [14]. IndeedBAS measures represent a usable tool to test subjective reward-sensitivity based on neurophysiological correlates [33,34,36,37,39,40,42,43,44,68]. Previous findings provide support for the role of Gray’s BAS in mediating approach behavior and dependence as associated with the drive to consume rewarding substances [14,69]. A direct association between the BAS and BAS subscales (BAS Drive, Fun Seeking and Reward Responsiveness) to substance abuse has been shown [70]. Indeed, it was shown that heightened BAS and drug addiction are related and the first may be considered predictive of substance abuse [71].

Secondly, the FRN modulations were correlated with reward-level (BAS and reward subscale of BAS). We underlined that a general and systematic FRN amplitude reduction in response to specific category can be interpreted as a reduced error monitoring of the behavior in response to that stimulus. The specificity of this feedback mechanism in marking the subject’s ability to correctly monitor their behavior may underline the higher BAS-level deficit in “reward prediction” for specific stimulus category. Similarly, BAS levels were also able to differentiate the P300 effect, since a strength association was found between the BAS increasing and the P300 high amplitude for specific rewarding category (gambling and videogames). This effect may be paralleled to the partial impairment of the attentional responsiveness to the less relevant (potentially less rewarding) cues and a converse disproportioned increase for more salient (potentially more rewarding) cues.

## 5. Conclusions

To summarize, the present findings indicate that individuals scoring very high on IAT attribute higher motivational salience to rewarding cues compared to more neutral conditions. This is reflected in the enhanced behavioral gaining (reduction of ERs and RTs), the reduction of the “error monitoring effect” for FRN and increased P300 amplitude for stimuli represented as more salient and rewarding. Our results may also support the idea that rewarding and attentional mechanisms, mediated respectively by FRN and P300 deflections, act as behavioral regulators during a decisional choice. Biases concerning feedback mechanisms were apparent in those individuals who extremely focalized on reward (reward bias) than individuals who did not base their decisions on reward. Thus, we propose to consider reward salience as an important aspect in decisional processes in subjects with high-IAT. Therefore, high sensitivity to IAT maybe considered as a marker of dysfunctional reward processing (reduction of monitoring) and cognitive control (higher attentional values) for specific cues. More generally a direct relationship among reward-related behavior, attentional bias, Internet addiction and BAS attitude may be suggested.

However some limitations may be adduced in the present research. Indeed firstly the nature of the sample (undergraduate students) may have introduced some “response bias” due to their specific profile in comparison with young adult or not university students. Therefore this aspect should be considered as a potential limitation to the generalizability of the present results and it should be taken into consideration for future research. Secondly, IAT may be integrated by other measures (for example behavioral measures) in future research, to improve the knowledge of AI profile. In addition, a more exhaustive analysis of the cortical localization of the two ERPs effects (FRN and P300) should be provided in future research by using also source localization. Moreover, other potential interesting ERPs effects should be provided, such as other specific attentional or emotional markers (i.e., N200 or Mismatch Negativity, MMN, or N400 effect, [72]. In addition, a more strength analysis should be conducted about the significance of the relationship between rewarding effect and BAS attitude, taking into account also the BIS measure as a potential marker of an antithetic behavior, that is a more “inhibitory” behavior (higher BIS) compared to a more reward-related behavior (higher-BAS) in internet addiction. Also, the significance of some sub-scales of BAS should be better explored, taking into account some contrasting results of previous research [73]. Finally, the intrinsic relationship between the cognitive performance (ERs and RTs in Go/NoGo performance) and the brain responsiveness should be tested, in order to verify the direct link between the cognitive outcomes and the functional significance of both FRN and P300.

## Figures and Tables

**Figure 1 brainsci-07-00081-f001:**
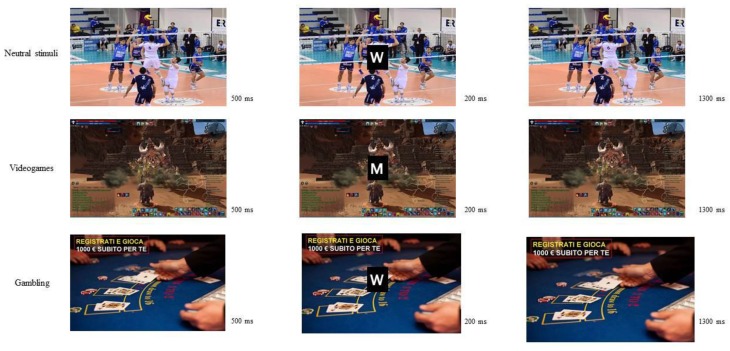
Experimental task. Each trial consisted of the presentation of a background picture (neutral, gambling and videogames) for 500 ms then the letter M or W appeared in the center of this picture for 200 ms.

**Figure 2 brainsci-07-00081-f002:**
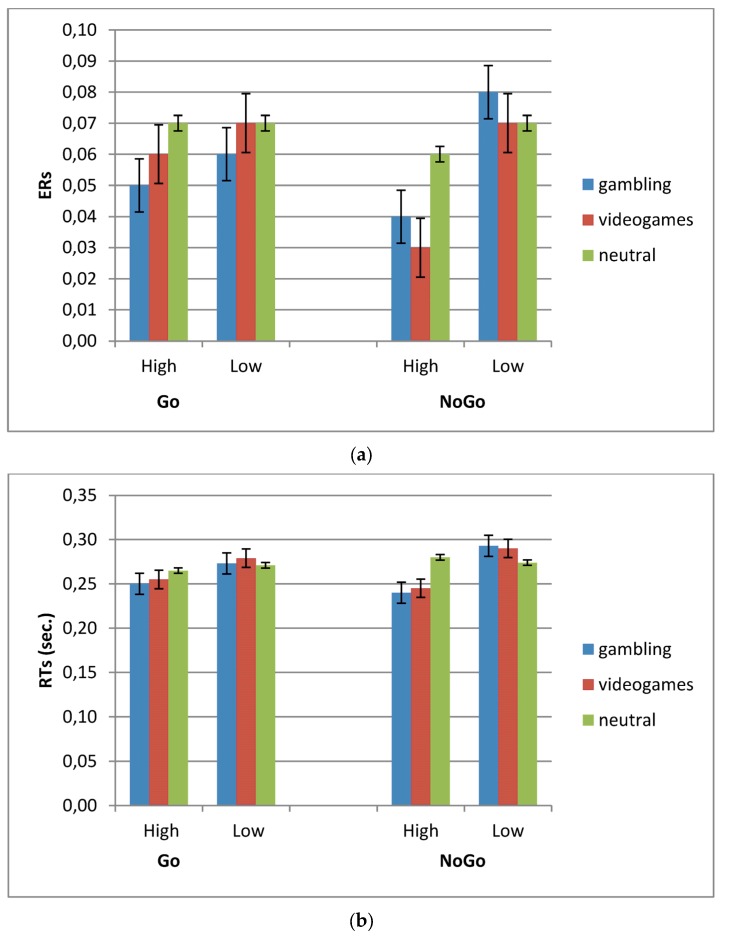
(**a**) ERs values as a function of stimulus type, Go/NoGo task and IAT; (**b**) RTs values as a function of stimulus type, Go/NoGo task and IAT.

**Figure 3 brainsci-07-00081-f003:**
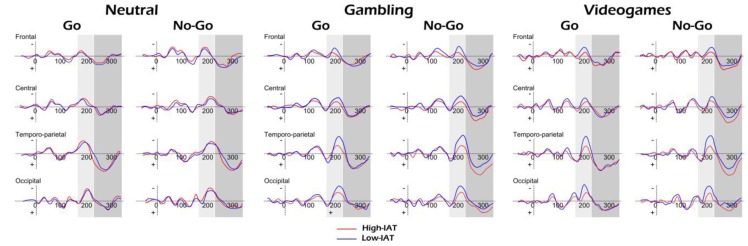
Mean peak ERP amplitude of FRN/P300 for high-BAS and low-BAS group.

**Figure 4 brainsci-07-00081-f004:**
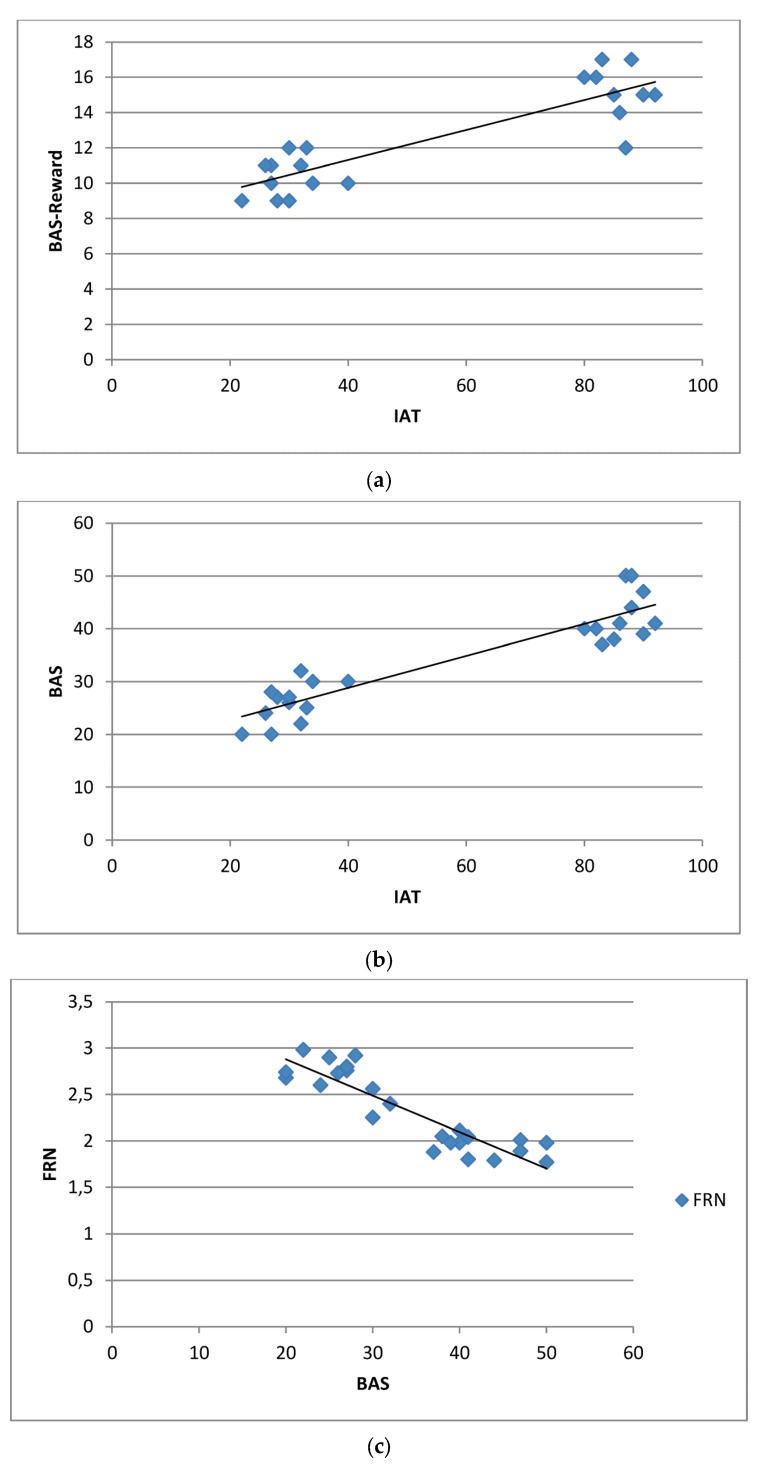
(**a**–**f**) Scatterplots of IAT, FRN/P300 and BAS (and BAS-Reward) Pearson correlations.

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
