# Peer review of "Evidences from Rewarding System, FRN and P300 Effect in Internet-Addiction in Young People"

_brainsci, 2017, doi:10.3390/brainsci7070081_

Round 1

Reviewer 1 Report

This is a very theoretically interesting article.  It makes a significant contribution to the field, while admittedly it remains a single explorative study and multiple replications are required.  I think there is lots of scope for improvement however.  Not least the language and grammar use is not acceptable, and beyond this, some of the phrasing is unclear and ambiguous and therefore unsuitable for publication.  Worse than this, I believe the writing style really buries the value of the findings observed.

Firstly, I found the tone of the report to be too informal for the journal.  I understand the tentative approach given the explorative nature of the research.  However, the informality of the writing style affects the reader comprehension and acceptance of the findings.  The tone could be interpreted as the author being highly tentative regarding the legitimacy and validity of the findings.  I’m sure that this is not reflective of the authors’ disposition.  To demonstrate, on page 1 line 39, states ‘A second main aspect implicated in IA is a deficit in the rewarding mechanism, able to induce a sort of ‘reward bias’ for potential rewarding cues.’  The tentativeness presented here suggests that there is theoretical ambiguity about what is being proposed.  I believe the tone of the article needs to be altered and presented more confidently (n.b. this does not mean being over-confident, presumptive etc, but rather demonstrate a clear set of propositions). 

Furthermore, I would recommend that the manuscript is thoroughly editing and corrected.  There a multiple of small errors throughout the manuscript.  For example, page 2 line 75 needs to be re-written, on page 3 line 95 VMPFC needs to come after its full title, page 3 line 115 needs to have appropriate spacing, page 13 line 393 there is an obvious spelling mistake.  There are a large amount of sentences that would benefit from re-editing and drafting to enhance clarity and reduce ambiguity.  Also there is a lack of engaging writing in the introduction, particularly regarding transitions.  It is not enough to start new points by continually re-using ‘About the X’, ‘About the Y’.

With respect the content, I think there are substantially important findings that have been presented, and they have been articulated very well within the results section.  However, I would hold contention regarding the proposition on page 2 line 91 and 92, where it is argued that insensitivity to punishment is particularly destructive.  Research demonstrates that a willingness to engage in compulsive and destructive behaviour can be increased in participants with high punishment sensitivity.  Essentially, losing or experiencing negative outcomes from a behaviour can overly stimulate discomfort where the participant is strongly motivated to quickly eliminate, such as losing in video-games, gambling activities etc.  I think this requires further clarification. See below article for example:

Gaher, R.M., Hahn, A.M., Shishido, H., Simons, J.S. & Gaster, S. (2015). Associations between sensitivity to punishment, sensitivity to reward and gambling. Addictive Behaviors, 42, 180–184.

I think further detail is required regarding the various methodological selections.  For example, the authors use the IAT developed by Young in the early days of the internet.  There have significant improvements in screening for internet dependence disorder.  There may be practical or theoretical arguments for using the IAT, and I think they need to be articulated to the reader.  Secondly, there is a similar need to elaborate on the correction used for ERP.  There are a multitude of possible corrections that could have been used, and potentially better corrections, therefore I think the authors need to justify the rationale for the corrections applied.  Furthermore, elaboration is required for how blinks were observed within the laboratory, and there is currently insufficient information to determine the potential validity of this solution.

Finally, I’m aware of the long history and often necessity of using undergraduates in such experiments.  However, I think it is critically important to discuss the potential representativeness of the findings, and concerns regarding applying to a non-undergraduate population.  For example, is not reasonable to assume that university students have higher levels of response inhibition, deferment of gratification and improved attention/evaluation, in comparison to non-undergraduate young adults.  I think this needs discussion and clarification.

Author Response

Reviewer 1

This is a very theoretically interesting article.  It makes a significant contribution to the field, while admittedly it remains a single explorative study and multiple replications are required.  I think there is lots of scope for improvement however.  Not least the language and grammar use is not acceptable, and beyond this, some of the phrasing is unclear and ambiguous and therefore unsuitable for publication.  Worse than this, I believe the writing style really buries the value of the findings observed.

Firstly, I found the tone of the report to be too informal for the journal.  I understand the tentative approach given the explorative nature of the research.  However, the informality of the writing style affects the reader comprehension and acceptance of the findings.  The tone could be interpreted as the author being highly tentative regarding the legitimacy and validity of the findings.  I’m sure that this is not reflective of the authors’ disposition.  To demonstrate, on page 1 line 39, states ‘A second main aspect implicated in IA is a deficit in the rewarding mechanism, able to induce a sort of ‘reward bias’ for potential rewarding cues.’  The tentativeness presented here suggests that there is theoretical ambiguity about what is being proposed.  I believe the tone of the article needs to be altered and presented more confidently (n.b. this does not mean being over-confident, presumptive etc, but rather demonstrate a clear set of propositions). 

Furthermore, I would recommend that the manuscript is thoroughly editing and corrected.  There a multiple of small errors throughout the manuscript.  For example, page 2 line 75 needs to be re-written, on page 3 line 95 VMPFC needs to come after its full title, page 3 line 115 needs to have appropriate spacing, page 13 line 393 there is an obvious spelling mistake.  There are a large amount of sentences that would benefit from re-editing and drafting to enhance clarity and reduce ambiguity.  Also there is a lack of engaging writing in the introduction, particularly regarding transitions.  It is not enough to start new points by continually re-using ‘About the X’, ‘About the Y’.

R: Thank you for this general suggestion. All the text, and specifically the Introduction section, was checked for clarity, correctness and expressions. The “informal” style was emended when necessary.

With respect the content, I think there are substantially important findings that have been presented, and they have been articulated very well within the results section.  However, I would hold contention regarding the proposition on page 2 line 91 and 92, where it is argued that insensitivity to punishment is particularly destructive.  Research demonstrates that a willingness to engage in compulsive and destructive behaviour can be increased in participants with high punishment sensitivity.  Essentially, losing or experiencing negative outcomes from a behaviour can overly stimulate discomfort where the participant is strongly motivated to quickly eliminate, such as losing in video-games, gambling activities etc.  I think this requires further clarification. See below article for example:

Gaher, R.M., Hahn, A.M., Shishido, H., Simons, J.S. & Gaster, S. (2015). Associations between sensitivity to punishment, sensitivity to reward and gambling. Addictive Behaviors, 42, 180–184.

R: Thank you for this important suggestion. This consideration was inserted and discussed, and the opportune reference was added.

I think further detail is required regarding the various methodological selections.  For example, the authors use the IAT developed by Young in the early days of the internet.  There have significant improvements in screening for internet dependence disorder.  There may be practical or theoretical arguments for using the IAT, and I think they need to be articulated to the reader.  Secondly, there is a similar need to elaborate on the correction used for ERP.  There are a multitude of possible corrections that could have been used, and potentially better corrections, therefore I think the authors need to justify the rationale for the corrections applied.  Furthermore, elaboration is required for how blinks were observed within the laboratory, and there is currently insufficient information to determine the potential validity of this solution.

Finally, I’m aware of the long history and often necessity of using undergraduates in such experiments.  However, I think it is critically important to discuss the potential representativeness of the findings, and concerns regarding applying to a non-undergraduate population.  For example, is not reasonable to assume that university students have higher levels of response inhibition, deferment of gratification and improved attention/evaluation, in comparison to non-undergraduate young adults.  I think this needs discussion and clarification.

R: The following aspects were integrated/modified, as requested:

-       IAT relevance was underlined within a specific paragraph (see method section)

-       Specific details were inserted about corrections procedure used for ERPs

-       Blinks control parameters was described

-       Limitation about the use of undergraduate university students as sample was inserted within the final paragraph

Reviewer 2 Report

The introduction section of the manuscript contains multiple paragraphs which begin with "about x, it was shown/it was observed." These openings should be edited for grammar and more dynamic English as it may sound repetitive and elementary in present form. Also consider editing down the length of the introduction.

The discussion section adequately demonstrates the experiments and reasoning behind the methods. In section 4.1 line 366, consider explaining more about the alternative to the  "immediate and impulsive" response, especially if a similar effect was observed for RTs. Why is it that subjective performance presented this way as opposed to an alternative? If there's no consensus on why this may be, discuss how this might relate to reward responses.

The conclusion may improve by expanding upon the limitations mentioned in line 461. Perhaps the IAT is not the best way to gauge internet addiction as it  consists only of a 20-item questionnaire with no behavioral observation. Of course, these are elements that can be considered for future studies. 

Minor grammatical changes should be made carefully throughout the manuscript, i.e. line 392 it was shown, not "showed."

Author Response

Reviewer 2

The introduction section of the manuscript contains multiple paragraphs which begin with "about x, it was shown/it was observed." These openings should be edited for grammar and more dynamic English as it may sound repetitive and elementary in present form. Also consider editing down the length of the introduction.

R. This aspect was considered (see also answer to similar request by Reviewer 1)

The discussion section adequately demonstrates the experiments and reasoning behind the methods. In section 4.1 line 366, consider explaining more about the alternative to the  "immediate and impulsive" response, especially if a similar effect was observed for RTs. Why is it that subjective performance presented this way as opposed to an alternative? If there's no consensus on why this may be, discuss how this might relate to reward responses.

R: Different interpretations were compared in a specific paragraph.

The conclusion may improve by expanding upon the limitations mentioned in line 461. Perhaps the IAT is not the best way to gauge internet addiction as it  consists only of a 20-item questionnaire with no behavioral observation. Of course, these are elements that can be considered for future studies. 

R: Possible limitations were inserted about IAT. However, as suggested by Reviewer 1, also the advantages in using this measurement were underlined (in method section).

Minor grammatical changes should be made carefully throughout the manuscript, i.e. line 392 it was shown, not "showed."

R: it was checked